**Data Availability Statement:** All relevant data are within the paper and its Supporting Information files.

**Funding:** B.H. received funding for this study from National Institutes of Health of USA. The funders had no role in study design, data collection and

# Regulation of host gene expression by J paramyxovirus

**Elizabeth R. Wrobel, Jared Jackson◉, Mathew Abraham¤, Biao He◉***

Department of Infectious Diseases, College of Veterinary Medicine, University of Georgia, Athens, Georgia, United States of America

¤ Current address: Merck & Co., Inc., West Point, Pennsylvania, United States of America
* bhe@uga.edu

## Abstract

Paramyxoviruses are negative-sense, single-stranded RNA viruses that are associated with numerous diseases in humans and animals. J paramyxovirus (JPV) was first isolated from moribund mice (*Mus musculus*) with hemorrhagic lung lesions in Australia in 1972. In 2016, JPV was classified into the newly established genus *Jeilongvirus*. Novel jeilongviruses are being discovered worldwide in wildlife populations. However, the effects of jeilongvirus infection on host gene expression remains uncharacterized. To address this, cellular RNA from JPV-infected mouse fibroblasts was collected at 2, 4, 8, 12, 16, 24, and 48 hours post-infection (hpi) and were sequenced using single-end 75 base pairs (SE75) sequencing chemistry on an Illumina NextSeq platform. Differentially expressed genes (DEGs) between the virus-infected replicates and mock replicates at each timepoint were identified using the Tophat2-Cufflinks-Cuffdiff protocol. At 2 hpi, 11 DEGs were identified in JPV-infected cells, while 1,837 DEGs were detected at 48 hpi. A GO analysis determined that the genes at the earlier timepoints were involved in interferon responses, while there was a shift towards genes that are involved in antigen processing and presentation processes at the later timepoints. At 48 hpi, a KEGG analysis revealed that many of the DEGs detected were involved in pathways that are important for immune responses. qRT-PCR verified that *Rtp4*, *Ifit3*, *Mx2*, and *Stat2* were all upregulated during JPV infection, while *G0s2* was downregulated. After JPV infection, the expression of inflammatory and antiviral factors in mouse fibroblasts changes significantly. This study provides crucial insight into the different arms of host immunity that mediate *Jeilongvirus* infection. Understanding the pathogenic mechanisms of *Jeilongvirus* will lead to better strategies for the prevention and control of potential diseases that may arise from this group of viruses.

## Introduction

Many important human and animal viruses exist within the *Paramyxoviridae* family. These viruses are negative-sense, singled-stranded RNA viruses that possess a nonsegmented genome. In 1972, J paramyxovirus (JPV) was isolated from moribund wild mice (*Mus musculus*) with hemorrhagic lung lesions in Australia [1]. In 2016, JPV was classified into the newly established genus *Jeilongvirus*. Seven other viruses have also been classified into this genus, all identified in rodents and bats [2]. Several more putative species have been reported in other

analysis, decision to publish, or preparation of the manuscript.

**Competing interests:** The authors have declared that no competing interests exist.

mammals, such as cats [3] and Belgian hedgehogs [4], indicating that jeilongviruses have a wider host range and may be more prevalent than previously thought. Jeilongviruses are distinct from other paramyxoviruses due to the presence of one or two extra genes that encode transmembrane proteins, between the fusion (F) and glycoprotein (G) genes. Further, viruses in the rodent subclade of *Jeilongvirus* have unusually large G genes and genome sizes [5].

Jeilongviruses have been detected in Asia [6–10], Africa [5, 11–16], Europe [4, 17, 18], and Central and South America [19]. Recently, Larsen et al. 2022 described 3 novel jeilongviruses from wild rodents in Arizona, USA [20]. New jeilongviruses continue to be identified in other countries. Two genetically distinct novel jeilongviruses, Paju Apodemus paramyxovirus 1 (PAPV-1) and 2 (PAPV-2), were discovered in mice trapped in the Republic of Korea [21]. PAPV-1 was able to infect human endothelial and epithelial cells. In Vanmechelen et al. 2022, various rodent and shrew species were tested for paramyxoviruses across Belgium and Guinea and the full-length genomes for 5 novel jeilongviruses were determined [22]. Wells et al. 2022 reported jeilongviruses sequences detected from bats sampled in Brazil and Malaysia [23]. In the Hainan Province of China, a novel jeilongvirus, termed HaParaV, was detected in 3.6% of bats sampled from two trapping sites [24].

The JPV genome is 18,954 nucleotides, which contains eights genes in the order 3'-N-P/V/ C-M-F-SH-TM-G-L-5' [25]. Two different strains of JPV exist: JPV-BH and JPV-LW. JPV-BH is highly pathogenic in mice, while JPV-LW causes no visible symptoms of disease. Li et al. 2013 showed that replacement of the L gene of JPV-BH with the L gene of JPV-LW caused the virus to be attenuated in mice, demonstrating the role of the L gene in viral pathogenesis [26]. The F protein of JPV is predicted to be a type I membrane protein. JPV G produces a putative 709 amino acid residue attachment protein that has a distal, putative second open reading frame (ORF), referred to at ORF-X, and has not been detected in infected cells. The G proteins binds cellular receptors while F proteins direct membrane fusion to allow virus entry into host cells [25]. JPV contains a small hydrophobic (SH) gene that is not present in all paramyxoviruses, and a unique transmembrane (TM) gene. The TM gene encodes for a type II glycosylated integral membrane protein that is 258 amino acids and promotes cell-to-cell fusion [27]. The SH gene encodes a 69 amino acid type I membrane protein, with a predicted N-terminal ectodomain of 5 residues, a transmembrane domain of 23 residues, and C-terminal cytoplasmic tail of 41 residues [25]. Mouse fibroblasts infected with a recombinant JPV virus that lacks the SH gene (JPVΔSH) produced significantly more TNF-α and underwent greater rates of apoptosis when compared to cells infected with JPV [28]. Further, TNF-α was increased in the serum levels of JPVΔSH-infected animals. JPVΔSH was attenuated in mice compared to the wild-type virus, indicating the SH protein plays a crucial role in virulence [28].

Previous studies have demonstrated that *Jeilongvirus* is widely distributed and infects various types of mammals, including bats. Thus, jeilongviruses have high zoonotic potential as bats are natural reservoirs of many zoonotic paramyxoviruses, such as Hendra and Nipah viruses [29]. However, there is little known about this new and emerging class of viruses. Given that JPV induces cytopathic effects *in vitro*, we hypothesize that JPV infection will induce significant changes in the cellular transcriptional response as the virus replicates, and that we will be able to quantify these differences using RNAseq. These results will provide us with a detailed overview of the changes in host gene expression caused by jeilongvirus infection.

## Materials and methods

### Cell culture and virus

Mouse fibroblast L929 cells (ATCC CCL-1) were maintained in Dulbecco's modified Eagle medium (DMEM) containing 10% fetal bovine serum (FBS) and 1% penicillin-streptomycin

(P/S) (Mediatech Inc., Manassas, VA). Vero cells (ATCC) cells were maintained in 5% FBS, 1% P/S DMEM. All cells were incubated at 37°C in 5% CO2. Cells were passed at a 1:4 dilution 1 day prior to use in order to achieve 80%-90% confluence upon infection.

As previously described, recombinant JPV-BH (JPV) was propagated from an early passage and sequenced to confirm that no mutations occurred [28]. Briefly, a T150 flask of 90–100% confluent Vero cells was infected at a multiplicity of infection (MOI) of 0.1 for 1 hour. The inoculum was then replaced with 2% FBS, 1% P/S DMEM and virus was propagated for 4–5 days. The viruses were collected after syncytia started to form. Plaque assays using Vero cells were performed to determine viral titers [28].

## Viral infection & RNA purification

One day before infection, L929 cells were split into sterile 12 well plates at a 1:4 dilution. The cells were infected with JPV at a MOI of 5 or mock-infected. The plates were incubated for 1 hour at 37°C and gently rocked every 15 minutes. The infection media was then replaced with fresh 10% DMEM, 1% P/S. Total RNA from cells was collected at 2, 4, 8, 12, 16, 24, and 48 hours post infection (hpi) using the QIAGEN RNeasy Mini Kit (catalog no. 74106, Qiagen). A Nanodrop© spectrophotometer (ND-1000 Nanodrop Technologies, Wilmington, DE, USA) was used to assess A260/A280 ratios for determination of initial RNA purity and concentration of each sample. Total cellular RNA was collected from 3 biological replicates per treatment group (JPV infection or mock infection) at each timepoint. All samples were stored at -80°C for subsequent analyses. Fig 1A depicts the JPV viral genome and a diagram of the experimental outline is shown in Fig 1B.

## RNA-seq library preparation and sequencing

All library preparation and sequencing were done at the Georgia Genomics and Bioinformatics Core (GGBC) at the University of Georgia (Athens, GA). Final RNA integrity of each replicate was assessed on an Agilent 2100 Bioanalyzer instrument (Model 62939B, Agilent

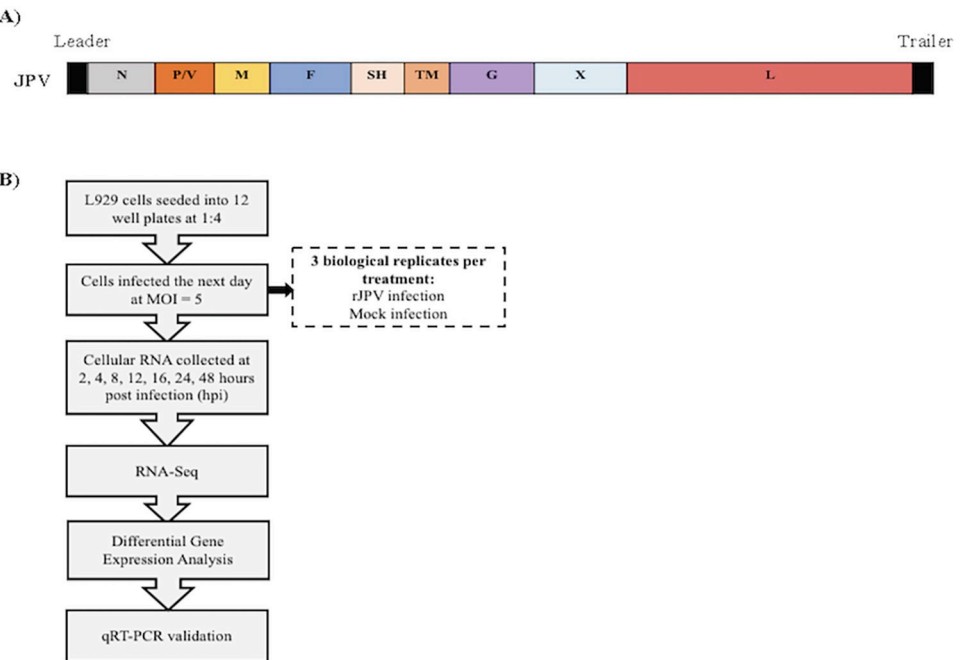

**Fig 1. Experimental outline.** (A) Schematic of JPV genome. (B) Overview of experimental set-up.

Technologies, Santa Clara, CA, USA). The KAPA Stranded mRNA-Seq kit was used to make a NGS stranded RNA library for each replicate (KK8421, KAPA Biosystems, Wilmington, MA, USA). The Roche LightCycler 480 II (product no. 05015278001, Roche Molecular Systems, Inc., Pleasanton, CA, USA) was used to pool all the libraries together by qPCR. For this step, the KAPA Library Quantification kit (Illumina) with qPCR Master Mix optimized for Light-Cycler 480 was used (KK4854, KAPA Biosystems, Wilmington, MA). Next, the quality of the pooled library was assessed before sequencing. The Qubit HS dsDNA assay (catalog no. Q32854, ThermoFisher Scientific, Waltham, MA, USA) was used to determine the DNA concentration of the pooled library. The Fragment Analyzer Automated CE System (Advanced Analytical Technologies, Ankeny, IA, USA) was used to visual the size distribution of the library. Then, qPCR was carried out and the PCR products were quantified as described above. The pooled libraries were sequenced on an Illumina NextSeq500 using a single-end 75 base pairs (SE75) high output flow cell. Each biological replicate was made up of 4 technical replicates.

### RNA-seq data processing and analysis

Computational work was done using the high-performance computing resources at the Georgia Advanced Computing Resource Center (GACRC) at the University of Georgia (Athens, GA). Trimmomatic software (version 0.36) [30] was used to remove adapters and quality trim raw reads. The phred quality score was set at 33. Trimmed reads below a 50-base threshold were discarded. Then, the reads from the 4 technical replicates for each biological replicate were combined into 1 file. Reads were assessed for quality using FastQC (version 0.11.8) both before and after trimming [31]. Next, the trimmed data was analyzed using the Tuxedo protocol described in Trapnell et al. 2012 [32]. TopHat2 2.1.1 [33] was used to align the trimmed reads for each merged biological replicate to the *Mus Musculus* reference genome (Ensembl GCA_000001635.8). The reference genome was indexed using Bowtie2 [34]. The reference transcriptome annotation was used in the TopHat2 code using the -G option. Raw reads, trimmed reads, and the percentage of reads that mapped to the reference genome for each biological replicate are reported in Tables 1 and 2. The mapped data was then processed by the Cufflinks package (version 2.2.1). First, transcripts for each replicate were assembled by passing the accepted_hit.bam file from Tophat2 through Cufflinks. Cufflinks outputted an assembly file for each sample. A single file that lists all of the assembly files was created and this was passed into Cuffmerge to create a single merged transcriptome annotation. The reference transcriptome annotation was included with Cuffmerge using the -g option. Then, Cuffdiff was run to quantify this merged transcriptome across the multiple conditions using the.bam files generated by the Tophat2 read alignments. A pairwise comparison between the JPV-infected replicates vs mock-infected replicates was made at each timepoint. Differentially expressed genes (DEGs) with a $|\log_2 FC|$ value $\geq 1$ with a false discovery rate (FDR) $\leq 0.05$ were considered statistically significant.

### Gene ontology (GO) enrichment and Kyoto Encyclopedia of Genes and Genomes (KEGG) pathway analysis

We used the GO database to identify significantly enriched biological processes, cellular components, and molecular functions using the PANTHER overrepresentation test [35–37]. For this analysis, the Fisher's exact test was performed with FDR correction.

The online Kyoto Encyclopedia of Genes and Genomes (KEGG) search tool was used to find the pathways that the 48 hpi DEGs were associated with using default parameters (https://www.genome.jp/kegg/mapper/search.html) [38].

## Validation of RNA-Seq data by real-time quantitative reverse transcription PCR (qRT-PCR) analysis

To verify the accuracy of the RNA-seq data, we measured mRNA expression of 5 DEGs at each timepoint using quantitative real-time PCR (qRT-PCR). PrimeTime® Predesigned qPCR primer-probes that are listed in S1 Table were used to measure expression of each gene (Integrated DNA Technologies, Coralville, IA, USA). The PCR assay was carried out using Taq-Path™ 1-Step RT-qPCR Master Mix, CG, (Applied Biosystems, Waltham, MA, USA) on a QuantStudio 3 Real-Time PCR System (Applied Biosystems, Waltham, MA, USA). Each 20µl reaction mixture had 1.5µl of the primer-probe, 5µl of RNA, 5µl of TaqPath™ 1-Step RT-qPCR Master Mix, and 8.5µl of ddH$_2$O. The PCR amplification protocol was as follows: 25˚C for 2 minutes, 50˚C for 15 minutes, 95˚C for 2 minutes, followed by 45 cycles at 95˚C for 3 seconds and 55˚C for 30 seconds. Each sample had 3 biological replicates and the *Mus musculus* glyceraldehyde-3-phosphate dehydrogenase (GAPDH) served as the housekeeping gene. The relative expression level of each gene was calculated using the $2^{-\Delta\Delta Ct}$ method and the error bars represent standard error of the mean (SEM). The correlation coefficient between the RNA-seq and qRT-PCR results for each gene was calculated using Graphpad Prism8 (v8.4.2, San Diego, CA, USA).

## Results

### Summary of sequencing data

Confluent L929 cells were infected with JPV at a MOI of 5. Total RNA was then extracted from the cells at 2, 4, 8, 12, 16, 24, and 48 hpi. Sequencing libraries were constructed for deep sequencing. The sequencing statistics for this study are provided in Tables 1 and 2. Using the

**Table 1. Summary of sequencing data for mock-infected biological replicates at each timepoint.**

| Mock infected cells | Raw Reads | Trimmed Reads | Mapped Reads | % Alignment |
|---|---|---|---|---|
| 2h_R1 | 21,680,237 | 21,296,863 | 20,346,192 | 95.5% |
| 2h_R2 | 24,311,313 | 23,849,790 | 22,114,960 | 92.7% |
| 2h_R3 | 21,020,090 | 20,602,353 | 19,108,631 | 92.7% |
| 4h_R1 | 25,821,413 | 25,323,105 | 24,161,199 | 95.4% |
| 4h_R2 | 26,123,236 | 25,658,858 | 24,039,798 | 93.7% |
| 4h_R3 | 31,504,306 | 30,903,955 | 28,951,357 | 93.7% |
| 8h_R1 | 22,520,071 | 22,112,993 | 21,084,283 | 95.3% |
| 8h_R2 | 28,493,848 | 27,967,607 | 26,572,365 | 95.0% |
| 8h_R3 | 33,059,187 | 32,401,010 | 30,197,962 | 93.2% |
| 12h_R1 | 28,544,554 | 28,018,198 | 25,998,195 | 92.8% |
| 12h_R2 | 38,183,470 | 37,349,362 | 34,938,032 | 93.5% |
| 12h_R3 | 30,566,957 | 29,978,326 | 27,896,746 | 93.1% |
| 16h_R1 | 31,234,301 | 30,657,885 | 29,088,737 | 94.9% |
| 16h_R2 | 56,254,912 | 55,216,916 | 52,366,052 | 94.8% |
| 16h_R3 | 27,581,414 | 27,075,548 | 25,682,134 | 94.9% |
| 24h_R1 | 21,344,316 | 21,280,916 | 20,625,766 | 96.9% |
| 24h_R2 | 28,529,218 | 28,035,968 | 26,759,687 | 95.4% |
| 24h_R3 | 25,723,694 | 25,237,096 | 24,046,804 | 95.3% |
| 48h_R1 | 57,475,835 | 56,591,953 | 53,939,441 | 95.3% |
| 48h_R2 | 59,284,478 | 58,221,363 | 55,313,114 | 95.0% |
| 48h_R3 | 51,455,687 | 50,588,427 | 48,065,853 | 95.0% |

**Table 2. Summary of sequencing data for JPV-infected biological replicates at each timepoint.**

| JPV infected cells | Raw Reads | Trimmed Reads | Mapped Reads | % Alignment |
|---|---|---|---|---|
| 2h_R1 | 19,406,755 | 19,040,378 | 17,520,146 | 92.0% |
| 2h_R2 | 21,828,967 | 21,403,007 | 19,764,974 | 92.3% |
| 2h_R3 | 24,034,292 | 23,559,988 | 21,623,175 | 91.8% |
| 4h_R1 | 30,518,047 | 29,912,878 | 26,673,112 | 89.2% |
| 4h_R2 | 26,111,302 | 25,605,605 | 22,937,756 | 89.6% |
| 4h_R3 | 25,961,807 | 25,476,521 | 22,962,641 | 90.1% |
| 8h_R1 | 28,772,220 | 28,215,323 | 24,291,152 | 86.1% |
| 8h_R2 | 38,962,890 | 38,211,272 | 32,914,813 | 86.1% |
| 8h_R3 | 34,723,428 | 34,033,054 | 29,336,467 | 86.2% |
| 12h_R1 | 16,814,961 | 16,451,156 | 14,218,104 | 86.4% |
| 12h_R2 | 31,155,888 | 30,576,997 | 26,535,372 | 86.8% |
| 12h_R3 | 23,056,872 | 22,638,559 | 20,021,269 | 88.4% |
| 16h_R1 | 31,658,167 | 31,084,945 | 28,062,088 | 90.3% |
| 16h_R2 | 13,706,148 | 13,444,192 | 12,063,032 | 89.7% |
| 16h_R3 | 37,956,343 | 37,311,184 | 33,719,543 | 90.4% |
| 24h_R1 | 31,450,045 | 30,845,229 | 25,622,271 | 83.1% |
| 24h_R2 | 28,496,193 | 27,957,740 | 23,881,930 | 85.4% |
| 24h_R3 | 31,450,657 | 30,851,071 | 26,571,732 | 86.1% |
| 48h_R1 | 50,926,797 | 50,101,915 | 38,262,945 | 76.4% |
| 48h_R2 | 68,392,299 | 67,314,633 | 51,189,425 | 76.0% |
| 48h_R3 | 50,333,217 | 49,599,964 | 38,107,920 | 76.8% |

Illumina NextSeq500 platform, 690,712,537 raw reads for mock-infected samples and 665,717,295 raw reads for JPV-infected samples were produced. The raw reads were trimmed to remove low-quality reads and reads with adaptor sequences. 678,368,492 clean reads were obtained for the mock-infected samples and 630,075,623 clean reads for the JPV-infected samples. The clean reads were mapped against the mouse (*Mus musculus*) reference genome. The percentage of reads that mapped to the reference genome varied by sample type and across the timepoints. The highest mapping percentage was 96.9% (24 hpi mock-infected sample, replicate 1). The lowest mapping percentage was 76.0% (48 hpi JPV-infected sample, replicate 2).

## Global changes in host gene expression during JPV infection

Following the Cuffdiff analyses, the number of DEGs ($|\log_2 FC| \geq 1$; FDR $\leq 0.05$) in the mock vs JPV comparison were identified for each timepoint (Table 3). Overall, the number of

**Table 3. The number of differentially expressed genes (DEGs) in the JPV-infected host cells.**

| Timepoint (hpi) | No. of DEGs |
|---|---|
| 2 | 11 |
| 4 | 94 |
| 8 | 426 |
| 12 | 718 |
| 16 | 830 |
| 24 | 1,071 |
| 48 | 1,837 |

DEG = $|\log_2 FC| \geq 1$; FDR $\leq 0.05$

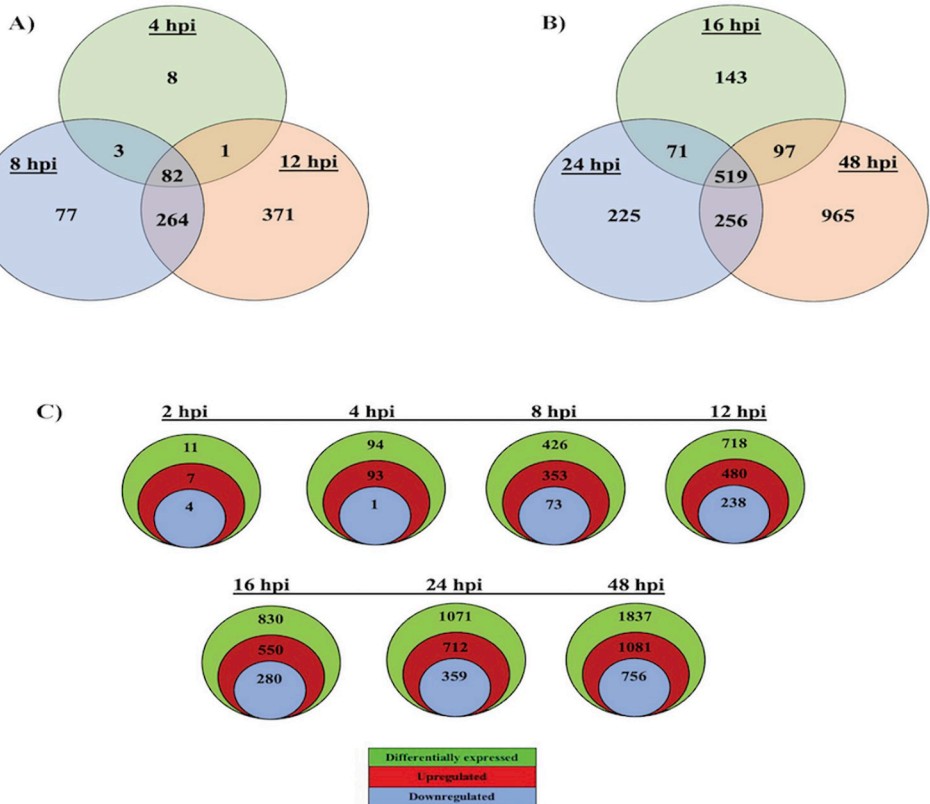

**Fig 2. Global gene expression data.** (A) Venn diagrams show an overlap of JPV-induced DEGs across 4, 8, and 12 hpi. (B) Venn diagrams show an overlap of JPV-induced DEGs across 16, 24, and 48 hpi. (C) The total number of upregulated and downregulated host genes in JPV-infected L929 cells.

significantly regulated genes increased over time. At 2 and 4 hpi, there were only 11 and 94 DEGs, respectively. At 8 hpi, there were 426 DEGs, and at 12 hpi, 718 DEGs. There was a slight increase 4 hours later, with 830 DEGs at 16 hpi. At 24 hpi, there were 1,071 DEGs detected. The final timepoint was 48 hpi, in which 1,837 DEGs were identified, demonstrating a dramatic increase in host gene activation as infection progressed. A Venn diagram revealed the shared or unique DEGs between 4, 8, and 12 hpi (Fig 2A). There were only 82 shared genes between the 3 earlier timepoints. There were 371 DEGs unique to 12 hpi. A Venn diagram shows that there were 519 shared genes between the later timepoints of 16, 24, and 48 hpi (Fig 2B). There were 965 DEGs unique at 48 hpi, demonstrating a large increase in host gene activation between 24 and 48 hpi. There are more up-regulated genes than down-regulated genes at each timepoint (Fig 2C).

At 2 hpi, there were only 11 DEGs, 7 of which are upregulated in the JPV-infected cells and 4 that are downregulated, as listed in Table 4. There are two immune-related genes being upregulated in the earliest stages of infection: chemokine (C-X-C motif) ligand 10 (*Cxcl10*) and chemokine (C-X-C motif) ligand 1 (*Cxcl1*). At 4 hpi, there was only 1 downregulated DEG among the 94 total DEGs detected. Interferon induced protein with tetratricopeptide repeats 1 (*Ifit1*), interferon induced protein with tetratricopeptide repeats 3 (*Ifit3*), and interferon induced protein with tetratricopeptide repeats 3B (*Ifit3b*) were among the top 10 upregulated DEGs at 4 hpi (S2 Table). Significant upregulation of Radical S-adenosyl methionine domain containing 2 (*Rsad2*) is also detected at 4 hpi, indicating that innate immune and antiviral

**Table 4. All upregulated and downregulated genes in JPV-infected cells at 2 hpi as ranked by fold change.**

| Upregulated | | | Downregulated | | |
|---|---|---|---|---|---|
| Gene Symbol | Gene Name | Log2FC | Gene Symbol | Gene Name | Log2FC |
| *Mir29b-2* | MicroRNA 29b-2 | +5.49 | *Jpx* | JPX transcript, XIST activator | -2.78 |
| *Tmem125* | Transmembrane protein 125 | +2.71 | *Sox2ot* | SOX2 overlapping transcript | -2.19 |
| *Cxcl10* | Chemokine (C-X-C motif) ligand 10 | +2.48 | *Ppp1r10* | Protein phosphatase 1 binding protein | -1.84 |
| *Stxbp2* | Syntaxin binding protein 2 | +1.87 | *Mir24-2* | MicroRNA 24–2 | -1.13 |
| *Gm37903* | – | +1.49 | | | |
| *Cxcl1* | Chemokine (C-X-C motif) ligand 1 | +1.24 | | | |
| *Apol7a* | Apolipoprotein L 7a | +1.09 | | | |

responses were being activated by this timepoint. Receptor transporter protein 4 (*Rtp4*) is upregulated at 4 hpi and remains highly upregulated at all following timepoints. At 8 hpi (S3 Table), DExD/H-Box helicase 60 (*Ddx60*) was significantly upregulated and remains activated at all later timepoints, and it is in the top 10 upregulated DEGs at both 24 and 48 hpi. Starting at 8 hpi, there was significant activation of multiple guanylate binding protein (*Gbp*) genes, such as *Gbp3*, *Gbp5*, *and Gbp2b*. At all following timepoints, there was at least one *Gbp* gene among the top 10 upregulated DEGs. *Gbps* are IFN-induced antiviral effectors that mediate innate immune responses [39]. A notable DEG among the top 10 upregulated genes at 12 hpi is tumor necrosis factor (ligand) superfamily, member 10 (*Tnfsf10*) (S4 Table). *Tnfsf10* is a cytokine belonging to the TNF family that contributes to immune surveillance against virus-infected cells by inducing apoptosis [40]. The top 10 upregulated DEGs between 16 hpi (S5 Table) and 24 hpi (S6 Table) are very similar. At 16 hpi, XIAP-associated factor 1 (*Xaf1*), an IFN-stimulated gene, is detected among the top 10. *Xaf1* is a proapoptotic ISG that also stabilizes interferon regulatory factor-1 (*IRF1*) to induce antiviral genes [41]. *Xaf1* is significantly upregulated in host cells infected with other emerging RNA viruses, such as Zika virus and SARS-CoV-2. The top 10 highly upregulated DEGs at 48 hpi are shown in Table 5.

**Table 5. Top 10 upregulated and downregulated genes in JPV-infected cells at 48 hpi as ranked by fold change.**

| Upregulate | | | Downregulated | | |
|---|---|---|---|---|---|
| Gene Symbol | Gene Name | Log2FC | Gene Symbol | Gene Name | Log2FC |
| *Ccl5* | Chemokine (C-C motif) ligand 5 | +10.95 | *Cyp4f40* | Cytochrome P450, family 4, subfamily f, polypeptide 40 | -6.23 |
| *Gbp5* | Guanylate binding protein 5 | +10.28 | *Sspo* | Scospondin | -5.99 |
| *Ddx60* | DExD/H-Box Helicase 60 | +10.17 | *Gdap1l1* | Ganglioside-induced differentiation-associated protein 1-like 1 | -5.60 |
| *Myo18b* | Myosin XVIIIB | +9.87 | *1700123O12Rik* | RIKEN cDNA 1700123O12 | -5.25 |
| *Ifit1* | Interferon induced protein with tetratricopeptide repeats 1 | +9.44 | *Aqp9* | Aquaporin 9 | -4.70 |
| *Rsad2* | Radical S-adenosyl methionine domain containing 2 | +9.20 | *Matn1* | Matrilin 1 | -4.48 |
| *Ifit3b* | Interferon induced protein with tetratricopeptide repeats 3B | +8.90 | *Igfbp5* | Insulin like growth factor binding protein 5 | -4.45 |
| *Mx1* | MX Dynamin Like GTPase 1 | +8.72 | *Ccdc7b* | Coiled-coil domain containing 7B | -4.43 |
| *Gas7* | Growth arrest specific 7 | +8.62 | *Kctd16* | Potassium channel tetramerization domain containing 16 | -4.42 |
| *Zbp1* | Z-DNA binding protein 1 | +8.19 | *Thsd7a* | Thrombospondin type 1 domain containing 7A | -4.38 |

## Gene ontology (GO) enrichment analysis

The GO database was used to estimate enriched biological processes, molecular functions, and cellular components for both the upregulated and downregulated genes sets at 4, 12, and 48 hpi. The upregulated DEGs at 4 hpi were mainly involved in biological processes related to the regulation of interferon responses, toll-like receptor signaling activity, and ribonuclease activity. The signaling pathways associated with MDA-5 and interleukin-27 (IL-27) were also highly enriched during early infection. For the molecular function category, the top 3 enriched functions were 2'-5'-oligoadenylate synthetase activity, adenylyltransferase activity, and double-stranded RNA binding. Host cell cytoplasm, symbiont-containing vacuole membrane, and host intracellular region were highly enriched under the category of cellular components (Fig 3A). At 12 hpi, there is still significant enrichment of interferon-related responses, particularly IFN-β, and regulation of ribonuclease activity (Fig 3B). However, the most significantly enriched biological process at 12 hpi was regulation of antigen processing and presentation via major histocompatibility complex (MHC) class I, indicating that genes involved in adaptive immune responses were being activated. This includes *Tap2*, which encodes a subunit of the transporter that delivers antigenic peptides to MHC class I proteins [42], and *Nod1*, which regulates the expression of MHC class I and II genes [43]. Almost all the enriched biological processes for upregulated DEGs at 48 hpi are related to antigen processing and presentation via MHC class I, demonstrating a continued increase in the activation of genes involved in T cell immunity since 12 hpi. There was also significant enrichment of the IFN-γ-mediated signaling pathway (Fig 3C). Shared enriched molecular functions between 12 hpi and 48 hpi included 2'-5'-oligoadenylate synthetase activity, TAP binding, beta-2-microglobulin binding, T cell receptor binding, CD8 receptor binding, and natural killer cell lectin-like receptor binding. Unique enriched molecular functions at 12 hpi included CXCR3 chemokine receptor binding and serine/threonine kinase binding, while phosphatidyl phospholipase B activity was only enriched at 48 hpi. The enriched cellular components at 12 hpi and 48 hpi are very similar to those identified at 4 hpi.

There were no significantly enriched GO terms for the downregulated DEGs at 4 hpi. The downregulated DEGs at 12 hpi were mainly involved in biological processes related to the regulation of syncytium formation by plasma membrane fusion, cellular response to lipoprotein particle stimulus, and positive regulation of receptor internalization (Fig 4A). The only enriched molecular GO terms at 12 hpi are protein binding and binding. No enriched cellular components were identified at this timepoint. At 48 hpi, the downregulated DEGs were connected to various glycolytic processes GO terms, such as the glucose catabolic process, canonical glycolysis, and the glycolytic process through both glucose-6-phosphate and fructose-6-phosphate (Fig 4B). The top 3 enriched molecular functions at 48 hpi included oxidoreductase activity, glucose binding, and fibronectin binding. Basement membrane, collagen-containing extracellular matrix, and mitochondrial membrane and envelope were all enriched under the cell composition category.

## KEGG enrichment analysis

The enriched DEGs at 48 hpi were analyzed using KEGG mapper (version 5.0) using the house mouse (*Mus musculus*, mmu) as a search model to identify canonical pathways. In this study, we identified 324 canonical pathways at 48 hpi, many of which have immunological functions. These canonical pathways include *NOD-like receptor signaling* (S1 Fig) and *toll-like receptor signaling* (S2 Fig). Additionally, the enriched canonical pathway for *RIG-I-like receptor signaling* is presented in S3 Fig and *TNF signaling pathway* in S4 Fig. The DEGs involved in the *PI3K-Akt signaling pathway* are also shown in S5 Fig. For these pathway maps, DEGs that

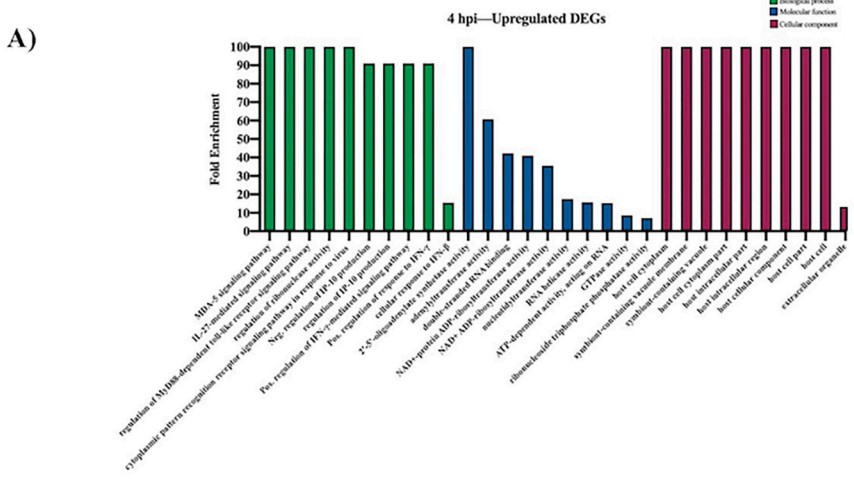

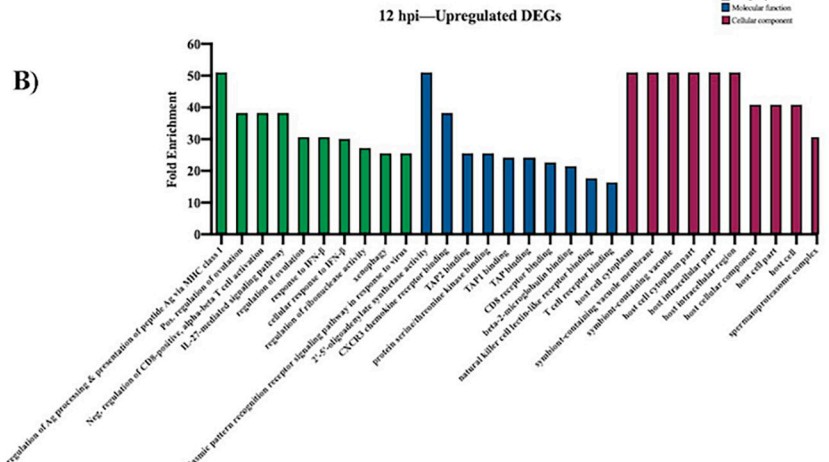

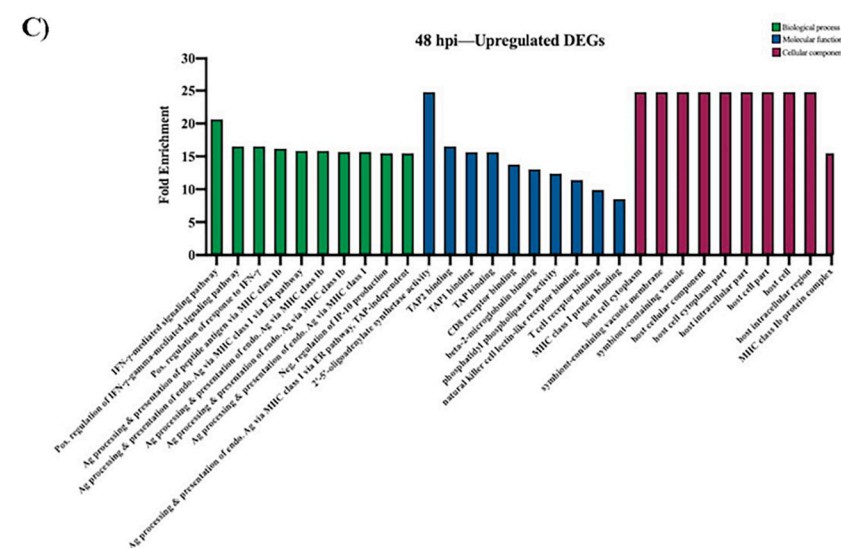

**Fig 3. Gene ontology analysis of upregulated DEGs during JPV infection (MOI = 5).** (A) GO analysis at 4 hpi, (B) 12 hpi, and (C) 48 hpi. The top 10 GO biological processes, molecular functions, and cellular components at each timepoint are listed.

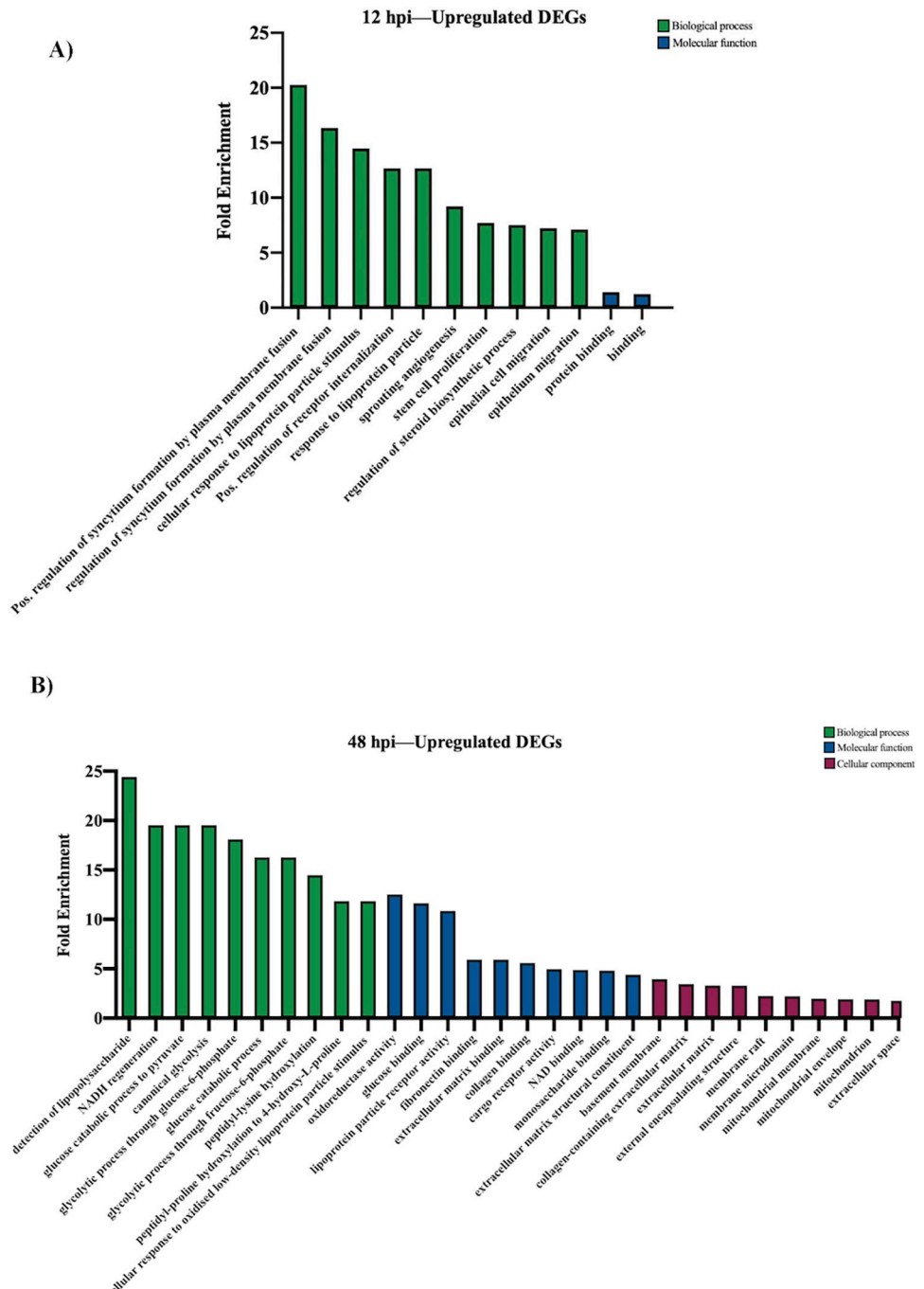

**Fig 4. Gene ontology (GO) analysis of downregulated DEGs during JPV infection (MOI = 5).** (A) GO analysis at 12 hpi and (B) 48 hpi. The top 10 GO biological processes, molecular functions, and cellular components at each timepoint are listed.

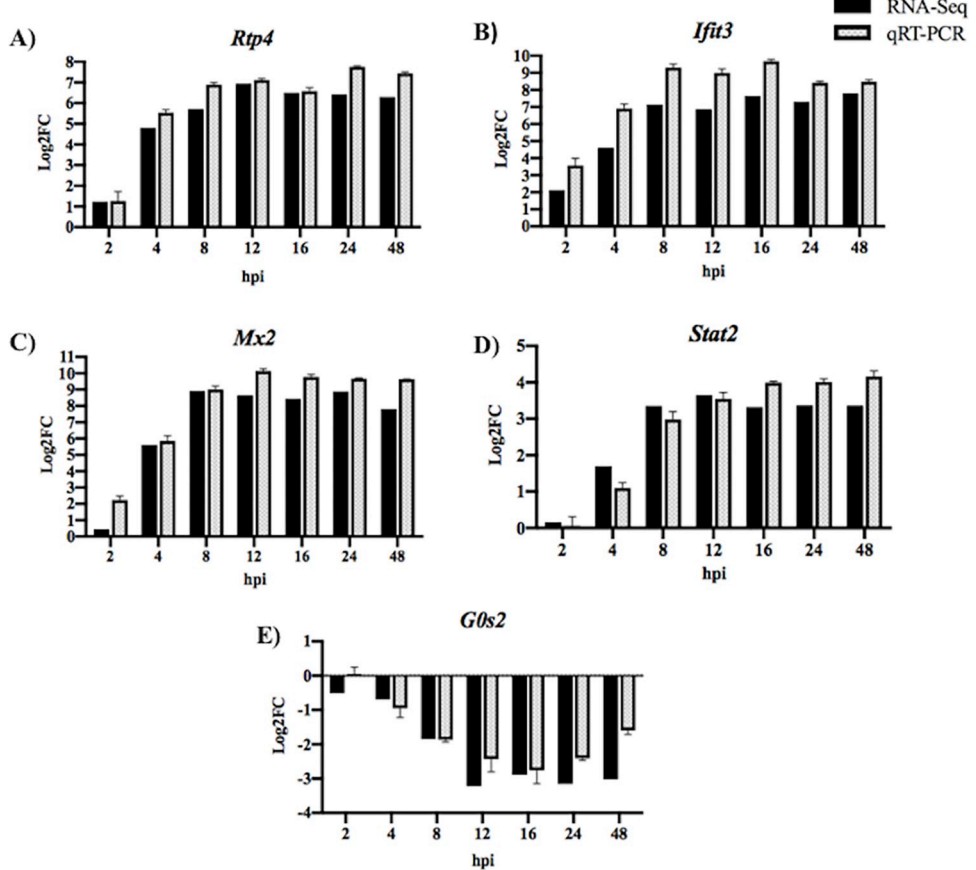

**Fig 5. Expression levels of host genes in L929 cells were verified using qRT-PCR.** To validate the expression results obtained from NGS, a subset of highly upregulated genes was selected for qRT-PCR analysis: (A) *Rtp4*, (B) *Ifit3*, (C) *Mx2*, and (D) *Stat2*. We also measured (E) *G0s2* expression, a gene that was significantly downregulated during infection. qRT-PCR was performed in triplicate and the error bars represent standard error of the mean (SEM).

were upregulated in the JPV-infected cells are indicated in red, while downregulated DEGs are blue. Genes that with no significant difference in expression between the virus-infected and mock cell are shaded green. Genes that had no detectable expression in either treatment group are shown in white.

## Validation of RNA-Seq data by real-time quantitative reverse transcription PCR (qRT-PCR)

To validate the RNA-seq data, a subset of 5 DEGs were selected for qRT-PCR analysis. As shown in Fig 5A–5E, the $\log_2$FC values for the 5 genes were very similar between the RNA-seq and qRT-PCR analysis. The correlation coefficient between the RNA-seq and qRT-PCR results for each gene was calculated (*Rtp4* = 0.9721; *Ifit3* = 0.9567; *Mx2* = 0.9736, *Stat2* = 0.9494, *G0s2* = 0.8751). These results support that the gene expression analysis obtained via RNA-seq is reliable.

## Discussion

The *Paramyxoviridae* family is in the order *Mononegavirales*, which is comprised of viruses that have a nonsegmented, negative-stranded RNA genome. *Jeilongvirus* is a recently proposed

genus of the *Paramyxoviridae* family, which includes rodent viruses such as JPV, Beilong virus (BeiPV), and Tailam virus (TlmPV). Jeilongviruses are unique from other paramyxoviruses in that they possess 1–2 additional membrane proteins, SH and TM, between the F and G genes [16]. Currently, there is very little known about jeilongviruses. However, surveillance studies are continuously discovering novel jeilongviruses in various types of mammals, including bats, rodents, and cats all over the world [2, 3].

Understanding the host immune response to *Jeilongvirus* is important for the control and prevention of potential diseases that may arise from this group of viruses. In recent years, transcriptome profiling has allowed researchers to generate a global view of the host genes and cellular pathways that are being activated or suppressed in response to viral pathogens [44]. Previous studies have used RNA-seq technology for transcriptome studies of other paramyxoviruses, such as Nipah virus [45], Newcastle disease virus [46], and Measles virus [47]. Our present study used L929 cells as an *in vitro* model to examine changes in host gene expression during JPV infection using RNA-seq. We selected seven timepoints to collect samples: 2, 4, 8, 12, 16, 24, and 48 hpi. This gave us a comprehensive understanding of the transcriptional changes in the host cells as JPV infection progressed. At the earliest timepoint of 2 hpi, there were only 11 DEGs detected. However, by 48 hpi, there were 1,837 DEGs (Table 3). A comparison of DEGs shared at the later timepoints revealed that there were 965 DEGs unique to 48 hpi, showing that the largest increase in gene expression levels occurred between 24 and 48 hpi (Fig 2B).

The highest upregulated DEGs at each timepoint were examined (Tables 4, 5 and S2–S6 Tables). Interestingly, *Cxcl1* is only detected at 2 hpi, while *Cxcl10* continues to be highly upregulated at all subsequent timepoints. *Cxcl10*, also known as *IP-10*, is secreted in response to IFN-γ and binds to the receptor CXCR3 receptor to induce chemotaxis, cell growth, apoptosis, and angiostasis [48]. The role of *Cxcl10* has been implicated to play a role in various types of viral infections, such as respiratory syncytial virus (RSV) [49] and dengue virus (DENV) [50]. This chemokine can either protect against or promote infection depending on the immune status and genetic background of the host [48]. Further studies are needed to examine if *Cxcl10* facilities or prohibits jeilongvirus infection. Receptor transporter protein 4 (*Rtp4*) is upregulated at 4 hpi and remains highly upregulated at all following timepoints. *Rtp4* is an interferon-stimulated gene (ISG) that demonstrates antiviral activity towards various flaviviruses, which are also RNA viruses [51, 52]. At 8 hpi, DExD/H-Box helicase 60 (*Ddx60*) was significantly upregulated and remains activated at all later timepoints and is in the top 10 upregulated DEGs at both 24 and 48 hpi. *Ddx60* has been proposed to function as a restriction factor highly specific for certain types of viruses [53]. For example, *Ddx60*-KO cells and mice showed no impairment in resistance to infection with various viruses, such as influenza A virus, encephalomyocarditis virus, Sindbis virus, vaccinia virus, or herpes simplex virus-1 [53]. In contrast, *Ddx60*-KO mice had increased mortality after infection with vesicular stomatitis virus [54] and *Ddx60* exhibited antiviral activity against hepatitis C virus (HCV) [51]. Interestingly, like JPV, both viruses possess an RNA genome.

Our GO analysis of upregulated DEGs revealed an increase in biological processes associated with immunological cell functions. At an earlier timepoint, 4 hpi, the highly enriched GO terms are heavily involved in pathogen recognition and innate immune signaling, such as *MDA-5 signaling pathway* and *MyD88-dependent toll-like receptor signaling pathway* (Fig 3A). At 48 hpi, the upregulated DEGs are associated with various biological processes that mediate adaptive immune responses, such as antigen processing and presentation of peptide antigen via MHC class Ib. Thus, during early JPV infection, there is high activation of genes associated with innate immunological processes, and as infection progresses, genes mediating adaptive responses become highly enriched (Fig 3B and 3C). Interestingly, downregulated DEGs at 48

hpi are associated with various glycolytic processes, suggesting that JPV may hijack this metabolic pathway to promote infection (Fig 4B). The metabolic pathway of glycolysis is an important host factor that can influence the outcome of viral infections and can be manipulated by some viruses to support infection [55]. For example, glycolysis acts as a virulence determinant during murine norovirus infection [56]. As JPV infection progressed, there was significant downregulation of genes involved in glycolysis, suggesting that jeilongvirus infection may influence this metabolic pathway. Additional studies will be needed to elucidate the exact role of glucose metabolism in the replication and pathogenesis of jeilongviruses.

There is immense diversity in the genomic features and replication dynamics of RNA viruses. Despite this, RNA viruses can induce common host cell components upon infection to trigger an innate immune response [57]. Our KEGG enrichment analysis of JPV-infected L929 cells at 48 hpi revealed the significant upregulation of genes enriched in antiviral immune pathways common to RNA viruses. Of these signaling pathways, pattern recognition receptors (PRRs), such as NOD-like receptors, toll-like receptors (TLRs), and RIG-I-like receptors, are essential for the recognition of viral RNA and subsequent initiation of the innate immune response through the activation of type I IFN and inflammatory cytokines [58]. At 48 hpi, the *Nod-like receptor signaling pathway* was highly enriched (S1 Fig). Both *Nod1* and *Nod2* were significantly upregulated at this timepoint, leading to the downstream activation of proinflammatory cytokines and *IFN-α/β*, which then stimulate various ISGs. Recent studies have uncovered the importance of NOD1 and NOD2 in sensing ssRNA viruses [59]. NOD1 regulates innate antiviral responses following HCV infection [60], while NOD2 knockdown results in increase viral titer of RSV [61] and influenza A virus [62]. Follow-up studies examining the potential roles of NOD1/2 in jeilongvirus infection are warranted. The *toll-like receptor signaling pathway* (S2 Fig) was highly enriched at 48 hpi, showing that *Tlr2* and *Tlr3* were upregulated during infection. TLRs are important for the detection of various pathogen-associated molecular pathogens (PAMPs) from bacteria, fungi, parasites, and viruses [59]. TLR2 plays an important role in controlling RNA viruses, such as RSV and measles virus [63, 64]. TLR3 mainly recognizes dsRNA and induces the activation of IRF3 and NF-kB, ultimately leading to the induction of type I IFNs [65]. TLR3 has been shown to recognize many RNA viruses including West Nile virus, poliovirus, and influenza A virus [66].

The transcription of innate immune molecules relies on activation of several transcriptional factors, such as interferon regulatory factors (IRFs) and signal transducers and activators of transcription (STATs). During JPV infection, expression of *Irf7* is upregulated. IRF7, which can be triggered through TLR signaling, is a crucial regulator of type I IFNs [67]. High expression of *Stat1* was also detected in the virus-infected cells. STAT1 is essential for the antiviral response by transducing in the nucleus the signal from type I, III, and III IFNs. Activation in *Stat1* also led to upregulation of the CXC chemokines interferon-gamma-inducible protein-10 (*IP-10*), monokine induced by interferon-gamma (*Mig*), and interferon-inducible T-cell alpha chemoattractant (*I-TAC*), all 3 of which exert chemotactic effects on T cells [68, 69]. Some key PRRs in the *RIG-I-like receptor signaling* pathway are activated during JPV infection, such as RIG-I and MDA5, which are encoded for by *Ddx58* and *Ifih1*, respectively (S3 Fig). RIG-I and MDA5 are cytoplasmic DEx(D/H) helicases that can detect intracellular viral products to trigger IFN-α/β production in infected cells [57]. RIG-I recognizes 5′-triphosphorylated or short RNA fragments, while MDA5 recognizes long RNA fragments or dsRNAs (not triphosphorylated) [58]. Signaling of RIG-I is triggered during infection by a variety of RNA viruses, including influenza virus, Japanese encephalitis virus, and other paramyxoviruses, including Newcastle disease virus (NDV) [70]. MDA5 is crucial for interferon production in response to RNA viruses such as picornaviruses [70]. For the TNF pathway (S4 Fig), there was significant

upregulation of downstream factors, including inflammatory cytokines interleukin-6 (*Il6*), interleukin-15 (*Il15*), and leukemia inhibitory factor (*Lif*).

The canonical pathway for PI3K-Akt signaling was enriched at 48 hpi (S5 Fig). This pathway is a critical regulator of various cellular processes, such as RNA processing, translation, autophagy, and apoptosis [71]. *Akt* expression is required by non-segmented, negative-sense RNA viruses to increase synthesis of viral RNA [72]. *Akt* was significantly upregulated during JPV infection at 48 hpi. Some viruses activate PI3K-Akt signaling as a strategy to support cell survival and viral growth. For example, activation of this pathway blocks apoptosis, thereby facilitating virus replication, assembly, and release [73]. Jeilongviruses may represent a class of new viruses that interfere with the host PI3K/Akt pathway to promote viral replication and survival. Further studies are needed to elucidate if any of the proteins encoded by JPV modulate PI3K/Akt signaling in the host. In this study, we have attempted to provide a comprehensive overview of the transcriptomic changes during JPV infection in L929 cells. A limitation of this study was the use a single immortalized mouse cell line. It is possible that the transformed cells have lost the ability to express certain genes that mediate JPV infection. Thus, future studies examining gene expression during JPV infection in both primary fibroblast cultures and *in vivo* are critical to obtain a complete understanding of the host-pathogen interactions of jeilongviruses.

In this study, the transcriptional levels of 5 DEGs were further analyzed by qRT-PCR (Fig 5). The $\log_2$FC values for these genes obtained by qRT-PCR were very similar to the values obtained by the RNAseq analysis, suggesting that our results were reliable. In conclusion, this study gives comprehensive insight into JPV-induced host responses *in vitro* and provides the foundation for further studies to examine the interactions between jeilongviruses and their host.

## Supporting information

**S1 Table. Primers used in qRT-PCR.**
(TIF)

**S2 Table. Top 10 upregulated and downregulated genes in JPV-infected cells at 4 hpi as ranked by fold change.**
(TIF)

**S3 Table. Top 10 upregulated and downregulated genes in JPV-infected cells at 8 hpi as ranked by fold change.**
(TIF)

**S4 Table. Top 10 upregulated and downregulated genes in JPV-infected cells at 12 hpi as ranked by fold change.**
(TIF)

**S5 Table. Top 10 upregulated and downregulated genes in JPV-infected cells at 16 hpi as ranked by fold change.**
(TIF)

**S6 Table. Top 10 upregulated and downregulated genes in JPV-infected cells at 24 hpi as ranked by fold change.**
(TIF)

**S1 Fig. The KEGG enriched canonical pathway for *NOD-like receptor signaling* at 48 hpi.** Red and blue shading indicates increased and decreased expression, respectively, in JPV-

infected cells relative to the mock-infected cells. White and green shading indicates non-expression and non-differential expression, respectively. Solid and dashed lines represent direct and indirect interactions, respectively.
(TIF)

**S2 Fig. The KEGG enriched canonical pathway for *toll-like receptor signaling* at 48 hpi.** Red and blue shading indicates increased and decreased expression, respectively, in JPV-infected cells relative to the mock-infected cells. White and green shading indicates non-expression and non-differential expression, respectively. Solid and dashed lines represent direct and indirect interactions, respectively.
(TIF)

**S3 Fig. The KEGG enriched canonical pathway for *RIG-I-like receptor signaling* at 48 hpi.** Red and blue shading indicates increased and decreased expression, respectively, in JPV-infected cells relative to the mock-infected cells. White and green shading indicates non-expression and non-differential expression, respectively. Solid and dashed lines represent direct and indirect interactions, respectively.
(TIF)

**S4 Fig. The KEGG enriched canonical pathway for *TNF signaling* at 48 hpi.** Red and blue shading indicates increased and decreased expression, respectively, in JPV-infected cells relative to the mock-infected cells. White and green shading indicates non-expression and non-differential expression, respectively. Solid and dashed lines represent direct and indirect interactions, respectively.
(TIF)

**S5 Fig. The enriched canonical pathway for PI3K-Akt signaling at 48 hpi.** Red and blue shading indicates increased and decreased expression, respectively, in rJPV-infected cells relative to the mock-infected cells. White and green shading indicates non-expression and non-differential expression, respectively. Solid and dashed lines represent direct and indirect interactions, respectively.
(TIF)

## Acknowledgments

We thank past and present members of Biao He's lab for technical support and helpful discussions. RNA-sequencing for this project was performed at the University of Georgia's Georgia Genomics & Bioinformatics Core (GGBC). Computational work was done using the high-performance computing resources at the Georgia Advanced Computing Resource Center (GACRC).

## Author Contributions

**Conceptualization:** Elizabeth R. Wrobel, Mathew Abraham, Biao He.

**Data curation:** Elizabeth R. Wrobel, Jared Jackson, Mathew Abraham.

**Formal analysis:** Elizabeth R. Wrobel, Mathew Abraham, Biao He.

**Funding acquisition:** Biao He.

**Investigation:** Elizabeth R. Wrobel, Jared Jackson, Mathew Abraham, Biao He.

**Methodology:** Elizabeth R. Wrobel, Jared Jackson, Mathew Abraham, Biao He.

**Project administration:** Elizabeth R. Wrobel, Biao He.

**Resources:** Biao He.

**Software:** Elizabeth R. Wrobel.

**Supervision:** Elizabeth R. Wrobel, Biao He.

**Validation:** Elizabeth R. Wrobel.

**Visualization:** Elizabeth R. Wrobel.

**Writing – original draft:** Elizabeth R. Wrobel, Biao He.

**Writing – review & editing:** Elizabeth R. Wrobel, Mathew Abraham, Biao He.

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
