## [Decision Letter · Decision Letter 0]

4 Jul 2023

PONE-D-23-13857Regulation of host gene expression by J paramyxovirusPLOS ONE

Dear Dr. He,

Thank you for submitting your manuscript to PLOS ONE. After careful consideration, we feel that it has merit but does not fully meet PLOS ONE’s publication criteria as it currently stands. Therefore, we invite you to submit a revised version of the manuscript that addresses the points raised during the review process.

We look forward to receiving your revised manuscript.

Kind regards,

Youkyung H. Choi, Ph.D.

Academic Editor

PLOS ONE

Journal Requirements:

   "This work was supported by NIH grant R01AI128924 to B.H."

5. Please upload a new copy of Figures 1-6 as the detail is not clear. Please follow the link for more information: https://blogs.plos.org/plos/2019/06/looking-good-tips-for-creating-your-plos-figures-graphics/" https://blogs.plos.org/plos/2019/06/looking-good-tips-for-creating-your-plos-figures-graphics/

6. Please upload a copy of Figure 7, to which you refer in your text on page 25. If the figure is no longer to be included as part of the submission please remove all reference to it within the text.

Reviewers' comments:

Reviewer's Responses to Questions

**Comments to the Author**

1. Is the manuscript technically sound, and do the data support the conclusions?

Reviewer #1: Yes

Reviewer #2: Yes

Reviewer #3: Yes

2. Has the statistical analysis been performed appropriately and rigorously? 

Reviewer #1: Yes

Reviewer #2: Yes

Reviewer #3: Yes

3. Have the authors made all data underlying the findings in their manuscript fully available?

Reviewer #1: Yes

Reviewer #2: Yes

Reviewer #3: No

4. Is the manuscript presented in an intelligible fashion and written in standard English?

Reviewer #1: Yes

Reviewer #2: Yes

Reviewer #3: Yes

5. Review Comments to the Author

Reviewer #1: This manuscript describes characterization of gene expression of mouse host cells upon infection with a jeilongvirus. In this study, the authors purified RNA from cells at various timepoints post infection and analyzed the gene expression using next gen sequencing, followed by Gene Ontology (GO) enrichment and Kyoto Encyclopedia of Genes and Genomes (KEGG) pathways analysis. All experimental conditions included biological and technical replicates. They identified multiple pathways that were either upregulated or downregulated during the first 48 hours post infection. As the time post infection increased, more pathways were either upregulated or downregulated.

Review notes:

1. This manuscript contains a lot of acronyms and jargon that are not always explained and/or spelled out. It would help

with the readability of this manuscript if this issue was resolved throughout the manuscript.

2. Line 65: define what F and G are.

3. Line 117: What was used for mock infection?

4. Line 119: What is P/S?

5. Line 147: Where is this GACRC located? At UGA?

6. Also, the figure labels are very small. There seems to be plenty of space to make the font larger and more readable.

Reviewer #2: In the manuscript “Regulation of host gene expression by J paramyxovirus”. Wrobel and colleagues have conducted time-course transcriptomic analyses after viral infection in mouse fibroblast, which would shed light on the prevention and control of potential diseases caused by J paramyxovirus. The manuscript is well written and the results are of general importance in immunity. I have some concerns before its acceptance.

1. Seven time points were designed after infection, while 48 h samples were more focused. I would suggest to combine all samples and concentrate on some common features, such as genes showing significant changes in all time points (or most time points). In addition, how about the determination of endpoint at 48 h, could longer duration lead to further immune responses?

2. There are some grammatical errors to be checked.

There should be space between number and hpi.

In the legend of Table 6. “Table 6” appeared for twice.

Reviewer #3: (1) The abstract has technical details (e.g. the name of the sequencer and the name of the software) that take up space that would be better used for more relevant information, like the length of reads and whether paired-end or not. The authors should be more explicit about the length of the reads and whether paired-end or not. It seems that “SE75” (line 143) means single end reads of 75 nt but that is not completely clear.

(2) The tables are a challenge of readers. Table 1 and Table 2 would be better placed with the Methods or as Supplementary Materials. This is important information but it is not directly pertinent to the main thrust of the report. The results in Table 3 could be presented in the text. Tables 4-10 repeat a lot of the same information. For example, why does Mx2 need to be defined in more than one table? The authors are advised to present this information in a more concise and readable form. Perhaps there could be summary table in main text and the extensive details for each time point in supplemenatry materialsl

(3) This was a single experiment with one strain of virus and one type of cell line. There could be more discussion of limitations of the study, such as the dependence on one lineage of mouse fibroblast cells. Transformed cells likely have lost the capacity for expression of some genes that would be relevant to infections of whole animals or primary cultures of fibroblasts. This does not mean that further experimentation is called for, but readers should be informed.

(4) Figure 5, which is large and just copies of pathways obtained from other sources. Much of this is not relevant here. Figure 5 could be deleted.

(5) The findings from this single experiment are important addition to the literature on this particular virus but were pretty much what would be expected in terms of responses to a RNA virus. The discussion could have devoted more space to a comparison with what has been found in other studies of virus infection of transformed fibroblast cells instead of dwelling so much on descriptions of what different genes and pathways do.

(6) I understand that this is a descriptive study without a clearly defined hypothesis but the authors could help the reader better understand what the bigger picture is here.

(7) I saw no mention of the public availability of the raw data, such deposit of the reads with SRA. There should also be an associated BioProject and BioSamples or the equivalent of these. This is essential.

6. PLOS authors have the option to publish the peer review history of their article (what does this mean?). If published, this will include your full peer review and any attached files.

Reviewer #1: No

Reviewer #2: No

Reviewer #3: **Yes: **Alan Barbour

---

## [Author Response · Author response to Decision Letter 0]

10 Aug 2023

Responses to Reviewers

Reviewer #1:

1. This manuscript contains a lot of acronyms and jargon that are not always explained and/or spelled out. It would help with the readability of this manuscript if this issue was resolved throughout the manuscript.

2. Line 65: define what F and G are. – 

Acronyms are now defined. Functions of F & G are further defined in lines 87-89.

3. Line 117: What was used for mock infection? 

L929 cells that were incubated with infection media that had no virus. After the 1hr incubation at 370C, the mock cells were washed with PBS and had maintenance media added back. This was the same treatment as the virus-infected cells. 

4. Line 119: What is P/S? 

1% penicillin-streptomycin (P/S), defined in lines 107-108 of the manuscript. 

5. Line 147: Where is this GACRC located? At UGA? 

Yes, it is at UGA – this was added in the manuscript at line 149. 

6. Also, the figure labels are very small. There seems to be plenty of space to make the font larger and more readable. 

The figure labels are sized based on PLOS ONE manuscript guidelines. 

Reviewer #2:

1. Seven time points were designed after infection, while 48 h samples were more focused. I would suggest combining all samples and concentrate on some common features, such as genes showing significant changes in all time points (or most time points). In addition, how about the determination of endpoint at 48 h, could longer duration lead to further immune responses? 

One purpose of this study was to examine differences in host gene expression across different timepoints during infection and to examine when certain host genes/pathways are triggered as JPV replicated. Thus, we feel it is important to keep data from each timepoint separated. However, in the revised manuscript, we have included a table that lists DEGs that are shared across most/all of the timepoints. When infecting with a high MOI, JPV titers reach the highest point around 48hpi. That is why we chose to examine changes in gene expression up until 48hpi because we wanted to see what genes are up- or down-regulated as viral replication increases exponentially. Further studies could examine changes in gene expression after 48hpi as viral titers decrease, but the differences in the host transcriptome may not be as pronounced. Alternatively, future studies could be done with a low MOI and examine differences in host gene expression with JPV replicating at a different rate. 

2. There are some grammatical errors to be checked.

There should be space between number and hpi. 

Fixed in the manuscript & in the figures. 

In the legend of Table 6. “Table 6” appeared for twice. 

Fixed 

Reviewer #3: 

The abstract has technical details (e.g. the name of the sequencer and the name of the software) that take up space that would be better used for more relevant information, like the length of reads and whether paired-end or not. The authors should be more explicit about the length of the reads and whether paired-end or not. It seems that “SE75” (line 143) means single end reads of 75 nt but that is not completely clear. 

SE75 means single-end 75 base paired sequencing, which is now denoted in the manuscript at line 146. We have now included this information in the abstract at line 38. 

The tables are a challenge of readers. Table 1 and Table 2 would be better placed with the Methods or as Supplementary Materials. This is important information but it is not directly pertinent to the main thrust of the report. The results in Table 3 could be presented in the text. Tables 4-10 repeat a lot of the same information. For example, why does Mx2 need to be defined in more than one table? The authors are advised to present this information in a more concise and readable form. Perhaps there could be summary table in main text and the extensive details for each time point in supplementary materials.

In these types of studies, the summary of sequencing data and the # of total DEGs detected at each timepoints is typically included in the results section. Thus, we feel it is important to keep Table 1, 2, and 3 in the results section. Table 3 is necessary for readers to easily see how many DEGs were found at each timepoint and show the dramatic increase in gene expression as infection progresses. The purpose of Tables 4-10 was to demonstrate at which timepoint certain immune-related genes started to be upregulated. We will include table 4 for 2hpi, and table 10 for 48hpi in the main manuscript to really highlight these differences and move the other tables to the supplementary materials. 

This was a single experiment with one strain of virus and one type of cell line. There could be more discussion of limitations of the study, such as the dependence on one lineage of mouse fibroblast cells. Transformed cells likely have lost the capacity for expression of some genes that would be relevant to infections of whole animals or primary cultures of fibroblasts. This does not mean that further experimentation is called for, but readers should be informed. 

Added in from lines 490-496: “In this study, we have attempted to provide a comprehensive overview of the transcriptomic changes during JPV infection in L929 cells. A limitation of this study was the use a single immortalized mouse cell line. It is possible that the transformed cells have lost the ability to express certain genes that mediate JPV infection. Thus, future studies examining gene expression during JPV infection in both primary fibroblast cultures and in vivo are critical to obtain a complete understanding of the host-pathogen interactions of jeilongviruses.”

Figure 5, which is large and just copies of pathways obtained from other sources. Much of this is not relevant here. Figure 5 could be deleted. 

Similar studies examining changes in cellular gene expression during viral infection have included these types of figures. These pathways were taken from the KEGG pathway database, which we used to show which genes in immune-related pathways are differentially expressed during JPV infection at 48 hpi. Thus, we feel it is important to include these pathways as a main figure of the paper, like other studies have done. 

The findings from this single experiment are important addition to the literature on this particular virus but were pretty much what would be expected in terms of responses to a RNA virus. The discussion could have devoted more space to a comparison with what has been found in other studies of virus infection of transformed fibroblast cells instead of dwelling so much on descriptions of what different genes and pathways do. 

We have updated the discussion section to compare what we found to other studies examining host genes/pathways in RNA viral infection. 

I understand that this is a descriptive study without a clearly defined hypothesis but the authors could help the reader better understand what the bigger picture is here. 

Added in lines 103-107 of manuscript: “Given that JPV induces cytopathic effects in vitro, we hypothesize that JPV infection will induce significant changes in the cellular transcriptional response as the virus replicates, and that we will be able to quantify these differences using RNAseq. These results will provide us with a detailed overview of the changes in host gene expression caused by jeilongvirus infection.”

I saw no mention of the public availability of the raw data, such deposit of the reads with SRA. There should also be an associated BioProject and BioSamples or the equivalent of these. This is essential.

The raw data have been already uploaded to NCBI SRA (sequence read archive) database https://www.ncbi.nlm.nih.gov/sra and will be publicly available on publication of our manuscript.

---

## [Decision Letter · Decision Letter 1]

3 Sep 2023

PONE-D-23-13857R1Regulation of host gene expression by J paramyxovirusPLOS ONE

Dear Dr. He,

Thank you for submitting your manuscript to PLOS ONE. After careful consideration, we feel that it has merit but does not fully meet PLOS ONE’s publication criteria as it currently stands. Therefore, we invite you to submit a revised version of the manuscript that addresses the points raised during the review process.

We invite you to submit a revised version of the manuscript that addresses the points below. Address the points raised by the reviewers, since one of them suggests that the Figures presented in the main text are not supported by experimental data, which is a critical criterion for future acceptance of the paper.  Please submit your revised manuscript by Oct 18 2023 11:59PM. If you will need more time than this to complete your revisions, please reply to this message or contact the journal office at plosone@plos.org. Please include the following items when submitting your revised manuscript:A rebuttal letter that responds to each point raised by the academic editor and reviewer(s). You should upload this letter as a separate file labeled 'Response to Reviewers'.A marked-up copy of your manuscript that highlights changes made to the original version. You should upload this as a separate file labeled 'Revised Manuscript with Track Changes'.An unmarked version of your revised paper without tracked changes. You should upload this as a separate file labeled 'Manuscript'.If applicable, we recommend that you deposit your laboratory protocols in protocols.io to enhance the reproducibility of your results. Protocols.io assigns your protocol its own identifier (DOI) so that it can be cited independently in the future. For instructions see: https://journals.plos.org/plosone/s/submission-guidelines#loc-laboratory-protocols. Additionally, PLOS ONE offers an option for publishing peer-reviewed Lab Protocol articles, which describe protocols hosted on protocols.io. Read more information on sharing protocols at https://plos.org/protocols?utm_medium=editorial-email&utm_source=authorletters&utm_campaign=protocols.

We look forward to receiving your revised manuscript.

Kind regards,

Youkyung H. Choi, Ph.D.

Academic Editor

PLOS ONE

Journal Requirements:

Additional Editor Comments:

We invite you to submit a revised version of the manuscript that addresses the points below. In particular, Figures and Tables presented in the main text are critical criteria for future acceptance of the paper.

Reviewers' comments:

Reviewer's Responses to Questions

**Comments to the Author**

1. If the authors have adequately addressed your comments raised in a previous round of review and you feel that this manuscript is now acceptable for publication, you may indicate that here to bypass the “Comments to the Author” section, enter your conflict of interest statement in the “Confidential to Editor” section, and submit your "Accept" recommendation.

Reviewer #2: (No Response)

Reviewer #3: (No Response)

2. Is the manuscript technically sound, and do the data support the conclusions?

Reviewer #2: Yes

Reviewer #3: Yes

3. Has the statistical analysis been performed appropriately and rigorously? 

Reviewer #2: Yes

Reviewer #3: Yes

4. Have the authors made all data underlying the findings in their manuscript fully available?

Reviewer #2: Yes

Reviewer #3: Yes

5. Is the manuscript presented in an intelligible fashion and written in standard English?

Reviewer #2: Yes

Reviewer #3: Yes

6. Review Comments to the Author

Reviewer #2: The authors state "in the revised manuscript, we have included a table that lists DEGs that are shared across most/all of the timepoints", could you specify which table as I did not find it?

Reviewer #3: The authors have submitted an improved manuscript. But I was not persuaded by the argument for including four figures of pathways. This may be done in other papers but that doesn't mean that it is justified here. There is just too little experimental data here for these sorts of conclusions. These figures may be okay in supplementary material but not in main article.

Minor point: There are too many significant figures in the tables. It is highly unlikely that the measurements were so accurate--out to severe decimal points.

7. PLOS authors have the option to publish the peer review history of their article (what does this mean?). If published, this will include your full peer review and any attached files.

Reviewer #2: No

Reviewer #3: No

---

## [Author Response · Author response to Decision Letter 1]

18 Oct 2023

Reviewer #2: 

The authors state "in the revised manuscript, we have included a table that lists DEGs that are shared across most/all of the timepoints", could you specify which table as I did not find it?

In place of that proposed table, we decided to include a table showing the top DEGs at 2 and 48hpi in the main manuscript. This will show the differences between the earliest and latest timepoint. The other timepoint tables have been moved to supplemental. We discuss commonly shared DEGs across the timepoints in the manuscript. 

Reviewer #3: 

The authors have submitted an improved manuscript. But I was not persuaded by the argument for including four figures of pathways. This may be done in other papers but that doesn't mean that it is justified here. There is just too little experimental data here for these sorts of conclusions. These figures may be okay in supplementary material but not in main article.

We have moved all pathway figures to the supplemental materials. We have removed 2 of the pathway figures. Now there are 5 pathway figures in supplemental instead of 7. 

Minor point: 

There are too many significant figures in the tables. It is highly unlikely that the measurements were so accurate--out to severe decimal points.

We changed to log2FC to have only 2 decimal points.

---

## [Editor Report · Decision Letter 2]

27 Oct 2023

Regulation of host gene expression by J paramyxovirus

PONE-D-23-13857R2

Dear Dr. He,

We’re pleased to inform you that your manuscript has been judged scientifically suitable for publication and will be formally accepted for publication once it meets all outstanding technical requirements.

Kind regards,

Youkyung H. Choi, Ph.D.

Academic Editor

PLOS ONE
---

## [Editor Report · Acceptance letter]

3 Nov 2023

PONE-D-23-13857R2 

Regulation of host gene expression by J paramyxovirus 

Dear Dr. He:

I'm pleased to inform you that your manuscript has been deemed suitable for publication in PLOS ONE. Congratulations! Your manuscript is now with our production department. 

Kind regards, 

on behalf of

Dr. Youkyung H. Choi 

Academic Editor

PLOS ONE